# Prevalence of High-Risk Human Papillomavirus Infections before and after Cervical Lesion Treatment, among Women Living with HIV

**DOI:** 10.3390/jcm10143133

**Published:** 2021-07-15

**Authors:** Timothée Dub, Sophie Le Cœur, Nicole Ngo-Giang-Huong, Wanmanee Matanasarawut, Pornnapa Suriyachai, Kannikar Saisawat, Chaiwat Putiyanun, Sudanee Buranabanjasatean, Prattana Leenasirimakul, Samreung Randaeng, Tristan Delory

**Affiliations:** 1Department of Health Security, Finnish Institute for Health and Welfare, 00160 Helsinki, Finland; 2Faculty of Associated Medical Sciences, Chiang Mai University, Chiang Mai 50200, Thailand; lecoeur@ined.fr (S.L.C.); Nicole.Ngo-Giang-Huong@phpt.org (N.N.-G.-H.); delory.tristan@gmail.com (T.D.); 3Institut National d’Etudes Démographiques (INED), 93322 Aubervilliers, France; 4Institut de Recherche pour le Développement (IRD) UMI 174-PHPT, 13002 Marseille, France; 5Ministry of Public Health, Lamphun Hospital, Lamphun 51000, Thailand; ogyn.lph@gmail.com; 6Ministry of Public Health, Phayao Provincial Hospital, Phayao 56000, Thailand; whan76_1@hotmail.com; 7Ministry of Public Health, Chiangrai Prachanukroh Hospital, ChiangRai 57000, Thailand; kannikarsai@gmail.com; 8Ministry of Public Health, Chiang Kham Hospital, Chiang Kham 56110, Thailand; putiyanun@hotmail.com; 9Ministry of Public Health, Mae Chan Hospital, Chiang Rai 57110, Thailand; sudanee2011@windowslive.com; 10Ministry of Public Health, Nakornping Hospital, Chiang Mai 50180, Thailand; leeprattana@gmail.com; 11Faculty of Medicine, Department of Pathology, Chiang Mai University, Chiang Mai 50200, Thailand; srangdae@mail.medicine.cmu.ac.th; 12Délégation à la Recherche Clinique et à l’Innovation (DRCI), Centre Hospitalier Annecy-Genevois, 74370 Epargny Mets-Tessy, France

**Keywords:** human papillomavirus, women living with HIV, cervical lesion

## Abstract

Even when receiving combination antiretroviral therapy, women living with HIV are at high risk of human papillomavirus (HPV) infection and/or cervical lesions, including cancer. Using data from the PapilloV prospective cohort, we evaluated the prevalence of high-risk HPV (HR-HPV) infections after cervical lesion treatment and investigated factors associated with their carriage. Women were followed up for three years with annual Pap smear and HPV genotyping. We offered treatment to women presenting either a Pap smear with high-grade squamous intraepithelial lesion or higher, and/or a biopsy showing cervical intraepithelial neoplasia II or III. We compared the prevalence of HR-HPV infection at the time of first treatment indication and at the end of follow-up among women who received treatment and those who did not. Overall, 46 women had treatment indication. HR-HPV prevalence significantly decreased from 67% to 27% (*p* value = 0.001) in the 30 women who received treatment, while it did not significantly decrease (from 56% to 38%) in the 16 women who did not (*p* value = 0.257). Due to lack of statistical power, the 40% relative difference in HR-HPV carriage between treated and untreated women was not significant. In women living with HIV, the treatment of a cervical lesion may be beneficial for clearing HR-HPV infections.

## 1. Introduction

Women living with HIV are at increased risk of presenting with a human papillomavirus (HPV) infection and/or cervical lesions, including cancer, even when receiving combination antiretroviral therapy [1,2,3]. 

In Thailand, among women living with HIV, the prevalence of cervical infection ranges from 25% to 62% for all types of HPV, and from 19% to 43% for high-risk HPV (HR-HPV) [4,5,6,7,8].

In a study evaluating the effect of cervical lesion treatment on HR-HPV carriage, women who had undergone punch biopsies had faster clearance of HR-HPV infections compared to the natural course of infection [9]. After treatment for cervical lesions, more women living with HIV had a persistent HR-HPV infection, compared to HIV-negative women [10]. In a larger study conducted in Nigeria, regardless of treatment, clearance of HR-HPV infection was similar (~8%) among HIV-negative and -positive women [11]. Independently of treatment, younger age was associated with faster HR-HPV clearance [12,13].

Among women living with HIV with a treatment indication for cervical lesion, we evaluated the impact of treatment on HPV carriage, including HR-HPV, between the time of cervical lesion detection and the end of follow-up, by comparing women who received treatment to those who did not. We also investigated potential factors associated with persistent carriage of HR-HPV.

We used data from the PapilloV cohort study, in which 829 women living with HIV in Thailand were screened annually for HPV infection and cervical lesions over three years. 

## 2. Materials and Methods

The PapilloV study was a prospective, multicenter cohort of women living with HIV, receiving antiretroviral therapy, nested in a larger HIV cohort in Thailand [14]. Women were recruited across 24 hospitals in Thailand. The study procedure has been detailed elsewhere [4]. 

In short, women were followed up for three years. Annual visits comprised a gynecological exam, including a Pap smear and HPV testing with genotyping. Colposcopy was performed if an abnormal Pap smear showed atypical squamous cells of undetermined significance (ASC-US) or higher-grade lesions and/or presence of either HR-HPV or potential HR-HPV (pHR-HPV) infections. Biopsy was performed in case of visible lesion at colposcopy.

Cervical lesion treatment was indicated in case of a Pap smear showing high-grade squamous intraepithelial lesion (HSIL) or higher and/or in case of a biopsy showing cervical intraepithelial neoplasia (CIN) II or III. Women underwent treatment as per physician decision. The study investigation team had no influence on whether treatment was offered or not to the study participants. 

HPV genotyping of cervical specimens was conducted using HPV-PapilloCheck (Greiner Bio-One, Frickenhausen, Germany). Genotypes considered high-risk HPV were the following: HPV 16, 18, 31, 33, 35, 39, 45, 51, 52, 56, 58, 59, and 68, while HPV 6, 11, 40, 42, 43, and 44/55 were considered as low-risk HPV genotypes. Due to the limited evidence regarding their potential carcinogenicity, the following genotypes (HPV 53, 66, 70, 73, and 82) were classified as potentially high-risk genotypes (pHR-HPV) [4]. Clinicians deciding for treatment indication were not made aware of the detailed HPV genotyping results but only of the presence of HR- or pHR-HPV.

The study protocol was approved by the ethics committees at the Institute for the Development of Human Research Protections, Ministry of Public Health, Thailand; the Faculty of Associated Medical Sciences, Chiang Mai University, Thailand; and at local hospitals, if established. The PapilloV study was registered at clinicaltrials.gov (NCT01792973).

In this analysis, we included women with cervical treatment indication, either at baseline or anytime during follow-up. We compared their characteristics at the time of treatment indication using the chi-squared test or the Fisher exact test (when required) for categorical variables and the Mann–Whitney U test for numerical values.

We compared the prevalence of each type of HPV infection (HR, pHR, and LR) at the time of the first treatment indication and at the end of follow-up among women who received treatment and among those who did not. Prevalence was compared using McNemar’s tests for paired nominal data (before vs. after in the same woman).

We investigated the possible characteristics associated with the presence of an HR-HPV infection by the end of the follow-up period, using bivariate logistic and exact logistic regressions adjusted on having received treatment. We selected the following variables a priori: age, CD4 cell count, HIV load, delay between ART initiation and cervical treatment indication, Nadir CD4 cell count, and cervical cytology at the end of the follow-up. Two time points were used in estimating prevalence: the time of treatment indication and at the end of follow-up. Hence, to be included, women had to have at least two HPV samples collected.

## 3. Results

From February 2012 to June 2013, 829 women were recruited in the cohort. Baseline characteristics, prevalence of HPV infections, and cervical lesions have been described elsewhere (4).

Over the median follow-up of 34.9 months (32.0 to 36.8) and the 2073 women-months of follow-up, 49 women had cervical lesions fulfilling treatment indication. Three of them were excluded from the analysis because of missing HPV samples either before or after treatment. The median duration between the initial and final HPV sample was 24.4 months [19.5 to 35.1]. Figure 1 describes the flow chart of women selection for this analysis. 

Among the 46 women included in the analysis, 14 (30%) had an HSIL on Pap smear, 12 (26%) had a CIN2, and 20 (43%) a CIN3 on the biopsy at the time of treatment indication. Thirty women received cervical treatment while 16 did not. Out of 30 women treated, 24 were treated once: among them, 22 experienced Loop Electrosurgical Excision Procedure (LEEP) and one participant a cold knife conization, while information on treatment type was missing for one participant. Among the six women who received two successive treatments: two had benefited from LEEP twice, while information was incomplete for four women. Apart from Pap smear and biopsy results, treated and untreated women had similar characteristics at the time of treatment indication (Table 1).

At the time of treatment indication, overall, 63% (29/46) of women had at least one HR-HPV infection. The most frequent HR-HPV genotypes were HPV 16 in 10 women (22%) and HPV 52 in seven women (15%), while only three women (7%) had an HPV-18 infection (Table 2 and Table 3).

Among women who received treatment, the prevalence of HR-HPV infection was 67% (IC_95%_, 48% to 82%) at the time of treatment indication and 27% (IC_95%_, 13% to 46%) at the end of follow-up, a 60% relative decrease of HR-HPV infection prevalence compared to the baseline (*p* value = 0.001) (Table 2). Among women who did not receive treatment, the HR-HPV prevalence was 56% (IC_95%_, 20% to 70%) at baseline versus 38% (IC_95%_, 16% to 65%) at the end of follow-up, a 32% relative decrease, which was not statistically significant (*p* value = 0.257) (Table 3).

There was no significant association between prevalence of HR-HPV infection by the end of the follow-up and the fact that women had received treatment (OR = 0.6 (IC_95_, 0.2 to 2.2), *p* value = 0.449).

Four women still had cervical lesions at the end of follow-up, two among those who received treatment (2 HSIL) (Appendix A
Table A1) and two among those who did not (1 HSIL and 1 CIN3) (Appendix A
Table A2). Details of HPV genotypes carriage by the end of follow-up are provided in Table 2 for treated women and in Table 3 for those untreated.

When adjusted for having received treatment, none of the women’s characteristics at the time of indication (age, CD4 cell count, HIV-RNA viral load, and antiretroviral therapy duration) were associated with HR-HPV carriage at the end of follow-up (Appendix A
Table A3). However, the presence of Pap smear showing ASC-US+ at the end of the follow-up was significantly associated with HR-HPV carriage: all HPV negative women had a normal Pap smear at the end of the follow-up (*p* value < 0.001).

## 4. Discussion

We found a significant decrease in HR-HPV carriage among women living with HIV who had been treated for a cervical lesion, while it was not significant among women who had not received treatment. However, the relative difference in decrease of HR-HPV infection prevalence was not significant between the two groups.

Interestingly, after adjusting for treatment, there was no significant association between women’s HIV characteristics and the presence of an HR-HPV infection at the end of the follow-up. However, the persistence of abnormal cytology findings (ASC-US+) was correlated with HR-HPV carriage by the end of the follow-up.

Some studies have assessed how treatment of cervical lesions might influence HPV clearance, but few have focused on women well-controlled for HIV infection. Also, it should be noted that HPV infections observed at the end of follow-up could be related to persistent infection, but also to new or recurrent infections. It was not possible to differentiate between them and this measurement bias cannot be addressed.

In 1997, Bollen et al. showed that HPV prevalence decreased from 98% to 31% when comparing samples before and after cervical treatment [15]. In that study, the five women living with HIV (not under antiretroviral therapy) all had positive HPV carriage after treatment. That study also indicated that detection of HPV infection after treatment was associated with incomplete conization. In our study, by the end of the follow-up, women with cervical abnormalities had a higher rate of HR-HPV infection, meaning that cervical lesions could be a proper medium for the persistence of HR-HPV infection. Also, in our study, all women were receiving effective antiretroviral therapy, and 97% had a CD4 level above 250 cells/mm^3^. This may have mitigated the mechanical effect of cervical treatment.

In a study in Nicaragua including 67 HIV-naïve women receiving LEEP for CIN2+ cervical lesions, 93% had at least one HR-HPV infection with genotypes 16, 31, 52, or 58 before treatment. Two years after treatment, the prevalence of HR-HPV infection was 8.4%, while it was 7.5% for CIN2+ cervical lesions [12]. In our study, the higher prevalence of HR-HPV carriage after treatment (27%) may be related to HIV infection, known to be a driver of incident HPV infections [1,2,3].

In a large study in China among HR-HPV infected women (HIV uninfected) without high-grade cervical lesions, viral clearance was estimated at 54.3% over a mean duration of 14.5 months, with higher clearance among women aged under 50 [13]. Evidence for age as a predictor of HPV clearance has been provided elsewhere [15]. This relationship reflects the natural course of HPV infection when ageing. In our study, probably due to lack of power, such association could not be evidenced.

Regardless of treatment, Travassos et al. showed a 46% clearance HR-HPV carriage after one year, among 62 women living with HIV and receiving antiretroviral therapy for more than a year [16]. This is consistent with the decrease we observed among women who received treatment (60%) and those who did not (30%). In our analysis, HPV 52 was the most frequent HR-HPV genotype evidenced both at the time of treatment indication and at the end of the follow-up. This high frequency was also evidenced among perinatally infected women in Thailand and Vietnam [7], emphasizing the need to scale-up nonavalent HPV vaccination programs in South-East Asia, at least among women living with HIV. The ongoing development of new L2-based vaccines may be promising for overcoming the diversity of oncogenic and potentially oncogenic HPV genotypes [17].

Our analysis is limited by a lack of statistical power, estimated post hoc at 16%. However, we did find that HR-HPV carriage decreased significantly over time among women who had received treatment.

## 5. Conclusions

Treatment of cervical lesions in women living with HIV under antiretroviral therapy may reduce the carriage of HR-HPV infections. Larger controlled cohort studies or meta-analyses are needed to assess which factors might be linked to a reduction of HR-HPV carriage among women with cervical lesions. In this population, nonavalent or new L2-based vaccination programs could be considered for preventing HPV-induced cervical cancer.

## Figures and Tables

**Figure 1 jcm-10-03133-f001:**
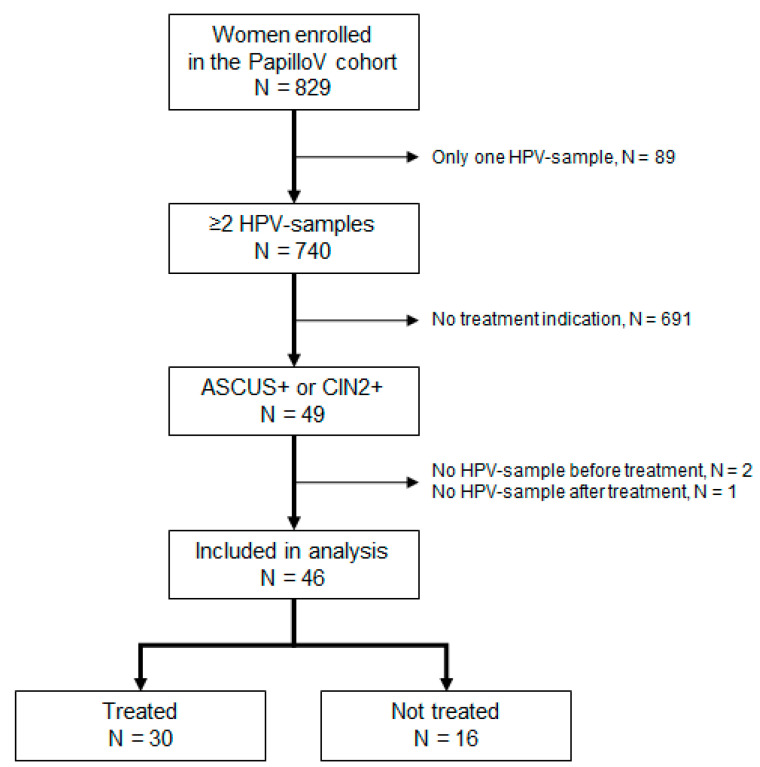
Flow chart of patient selection.

**Table 1 jcm-10-03133-t001:** Baseline characteristics at the time of indication among women who received treatment and women who did not.

	Received Treatment (*n* = 30)	Did Not Receive Treatment (*n* = 16)	*p* Value
	n (%), Median (IQR)	n (%), Median (IQR)	
Age (years) *	39.5 (33.1–43.7)	39.4 (35.4–42.5)	0.890
CD4 cell count			
>250 cells/mm^3^	29 (97)	15 (94)	1.000
<250 cells/mm^3^	1 (3)	1 (6)	
HIV load			
<1.7 log10 copies/mL	27 (90)	15 (94)	1.000
>1.7 log10 copies/mL	3 (10)	1 (6)	
Time since ART initiation (years) *	2.2 (1.0–4.8)	3.2 (2.2–4.4)	0.181
HPV infections			
HR-HPV infections	20 (67)	9 (56)	0.534
pHR-HPV infections	3 (10)	3 (19)	0.405
LR-HPV infections	3 (10)	2 (13)	1.000
Any HPV infection	23 (77)	11 (69)	0.726
Multiple HPV infection	6 (20)	4 (25)	0.720
Biopsy results			
CIN 2 –	1 (3)	8 (73)	<0.001
CIN 2 +	29 (97)	3 (27)	
Pap smear results			
HSIL –	20 (69)	3 (19)	0.002
HSIL +	9 (31)	13 (81)	
Combination of biopsy and Pap smear results			
HSIL +/CIN2 +	8 (28)	0	<0.001
HSIL –/CIN2 +	20 (69)	3 (27)	
HSIL +/CIN2 –	1 (3)	8 (73)	
HSIL –/CIN2 –	0	0	
Duration (months) between treatment indication and the last visit *	24.4 (14.0–34.6]	26.7 (20.5–35.1)	0.356

* Median (IQR), Mann–Whitney test.

**Table 2 jcm-10-03133-t002:** Prevalence of HPV infection at the time of indication and by the end of the follow-up, among women who received treatment for a cervical lesion.

		Before Treatment Indication	End of Follow-Up	*p* Value
	Genotypes	(*n* = 30)		
HIGH-RISK HPV INFECTIONS	HPV16	6 (20%)	1 (3%)	0.059
	HPV 18	1 (3%)	0 (0%)	0.317
	HPV 31	3 (10%)	0 (0%)	0.083
	HPV 33	3 (10%)	1 (3%)	0.157
	HPV 35	0	0 (0%)	-
	HPV 39	3 (10%)	0 (0%)	0.083
	HPV 45	1 (3%)	1 (3%)	1.000
	HPV 51	1 (3%)	0 (0%)	0.317
	HPV 52	5 (17%)	4 (13%)	0.564
	HPV 56	1 (3%)	1 (3%)	1.000
	HPV 58	1 (3%)	0 (0%)	0.317
	HPV 59	1 (3%)	0 (0%)	0.317
	HPV 68	2 (7%)	0 (0%)	0.157
	HR-HPV regrouped	20 (67%)	8 (28%)	0.001
POTENTIALLY HIGH-RISK HPV INFECTIONS	HPV 53	1 (3%)	0 (0%)	0.317
	HPV 66	1 (3%)	0 (0%)	0.317
	HPV 70	1 (3%)	1 (3%)	1.000
	HPV 73	0 (0%)	0 (0%)	-
	HPV 82	0 (0%)	0 (0%)	-
	pHR-HPV regrouped	3 (10%)	1 (3%)	0.317
LOW-RISK HPV INFECTIONS	HPV 6	0 (0%)	0 (0%)	-
	HPV 11	0 (0%)	0 (0%)	-
	HPV 40	0 (0%)	0 (0%)	-
	HPV 42	1 (3%)	0 (0%)	0.317
	HPV 43	0 (0%)	0 (0%)	-
	HPV 44/55	2 (7%)	0 (0%)	0.157
	LR-HPV regrouped	3 (10%)	0 (0%)	0.083
Any HPV infection		23 (77%)	9 (30%)	<0.001

**Table 3 jcm-10-03133-t003:** Prevalence of HPV infection at the time of indication and by the end of the follow-up among women who did not receive treatment for cervical lesion.

		Before Treatment Indication	End of Follow-Up	*p* Value
	Genotypes	(*n* = 16)		
HIGH-RISK HPV INFECTIONS	HPV16	4 (25%)	1 (6%)	0.083
	HPV 18	2 (13%)	1 (6%)	0.317
	HPV 31	1 (6%)	1 (6%)	-
	HPV 33	1 (6%)	1 (6%)	-
	HPV 35	0 (0%)	0 (0%)	-
	HPV 39	0 (0%)	0 (0%)	-
	HPV 45	0 (0%)	0 (0%)	-
	HPV 51	1 (6%)	2 (13%)	0.317
	HPV 52	2 (13%)	2 (13%)	1.000
	HPV 56	0 (0%)	0 (0%)	-
	HPV 58	0 (0%)	1 (6%)	0.317
	HPV 59	0 (0%)	0 (0%)	-
	HPV 68	0 (0%)	1 (6%)	0.317
	HR-HPV regrouped	9 (56%)	6 (38%)	0.257
POTENTIALLY HIGH-RISK HPV INFECTIONS	HPV 53	1 (6%)	0 (0%)	0.317
	HPV 66	0 (0%)	0 (0%)	-
	HPV 70	1 (6%)	1 (6%)	1.000
	HPV 73	1 (6%)	0 (0%)	0.317
	HPV 82	0 (0%)	0 (0%)	-
	pHR-HPV regrouped	3 (19%)	1 (6%)	0.157
LOW-RISK HPV INFECTIONS	HPV 6	0 (0%)	0 (0%)	-
	HPV 11	2 (13%)	1 (6%)	0.317
	HPV 40	0 (0%)	0 (0%)	-
	HPV 42	0 (0%)	0 (0%)	-
	HPV 43	1 (6%)	0 (0%)	0.317
	HPV 44/55	1 (6%)	1 (6%)	-
	LR-HPV regrouped	2 (13%)	2 (13%)	-
Any HPV infection		11 (69)	6 (38%)	0.059

## Data Availability

The data presented in this study are available on request from the corresponding author. The data cannot be made publicly available due to patients’ confidentiality.

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
