# Peer review of "Prevalence of High-Risk Human Papillomavirus Infections before and after Cervical Lesion Treatment, among Women Living with HIV"

_jcm, 2021, doi:10.3390/jcm10143133_

Round 1

Reviewer 1 Report

The article by Dub et al., describes the HPV prevalence before and after treatment in HIV positive patients in Thailand. Considering the sample size, outcome of the result, and no new information presented, it is of limited geographical importance. It adds a limited value if it is a first study in Thailand, or else, article fails to add any new information.

  1. Authors should specifically mention why the ‘untreated control group’ did not receive any treatments. Even though such a group is essential from experimental view, ethically it is disturbing to know why this group did not receive any treatment? Is it voluntarily left untreated because of patient’s choice despite they were given the option of treatment? Please explain.
  2. In M&M section, please include the genotypes of LR-HPV tested, which is missing.
  3. Please explain what type of treatment was given to “treated” group. Is there any correlation between HPV prevalence to treatment after the treatment? i.e. 28% that showed positive were they received same treatment as the 62% that responded to treatment?

Minor corrections –

  1. Please align table 1 properly
  2. Table 3 title has to be written above the table.
  3. Table 3 LR-HPV table contents don’t add-up, please correct.
  4. Replace “Viral load” as “HIV load” in table 1.

Author Response

The article by Dub et al., describes the HPV prevalence before and after treatment in HIV positive patients in Thailand. Considering the sample size, outcome of the result, and no new information presented, it is of limited geographical importance. It adds a limited value if it is a first study in Thailand, or else, article fails to add any new information.

We thank reviewer 1 for their assessment of our work and hope that we answer to all their comments.

  1. Authors should specifically mention why the ‘untreated control group’ did not receive any treatments. Even though such a group is essential from experimental view, ethically it is disturbing to know why this group did not receive any treatment? Is it voluntarily left untreated because of patient’s choice despite they were given the option of treatment? Please explain.
    Treatment was conducted at the discretion of the treating physician. This information is mentioned in the third paragraph of the Material and Methods section (line 97). We also added a sentence explaining that the study team had no influence on whether women would receive treatment or not (lines 98-99).
    The study investigation team had no influence on whether treatment was offered or not to the study participants.”.
  2. In M&M section, please include the genotypes of LR-HPV tested, which is missing.
    We thank the reviewer for noticing that this information was missing. We have now added it to the Material and Methods section (lines 101 to 107).
    Genotypes considered high-risk HPV were the following: HPV 16, 18, 31, 33, 35, 39, 45, 51, 52, 56, 58, 59, and 68, while HPV 6, 11, 40, 42, 43 and 44/55 were considered as low-risk HPV genotypes. Due to the limited evidence regarding their potential carcinogenicity, the following genotypes (HPV 53, 66, 70, 73, and 82) were classified as potentially high-risk genotypes (pHR-HPV)
  3. Please explain what type of treatment was given to “treated” group. Is there any correlation between HPV prevalence to treatment after the treatment? i.e. 28% that showed positive were they received same treatment as the 62% that responded to treatment?
    Following this comment, we added additional information regarding the treatment received in the results section (lines 151 to 157). However, as a majority of women benefited from LEEP, we decided not to compare whether post-treatment prevalence differed depending on the treatment received.
    “Out of 30 women treated, 24 were treated once: among them, 22 experienced Loop Electrosurgical Excision Procedure (LEEP) and one participant a cold knife conization, while information on treatment type was missing for one participant. Among the six women who received two successive treatments: two had benefited from LEEP twice, while information was incomplete for four women.”

Minor corrections –

  1. Please align table 1 properly
    Tabulations were removed from the first column of the table
  2. Table 3 title has to be written above the table.
    This was corrected
  3. Table 3 LR-HPV table contents don’t add-up, please correct.
    It is expected that totals do not add up in Table 3 due to possible multiple infections. For low-risk HPV infection at the time of treatment indication, one woman was infected with HPV 11, HPV 43 and 44/55, while one woman was only infected with HPV11, hence total of women infected with any low-risk HPV infection at the time of treatment indication was 2.

  4. Replace “Viral load” as “HIV load” in table 1.
    This was corrected in Table 1

Reviewer 2 Report

Interesting article assessing HPV natural history following among women with and without treatment for cervical pre-cancer among women with HIV. There are some limitations with how the data is currently being presented and questions on what statistical methods were used for analyses.

Major

  1. Table 2 & 3 categories need to be updated to include more columns on the natural history of the infection. An HPV infection or a lesion can clear or persist. We can also have a new (incident) HPV infection. “Before treatment” need to be labeled as a prevalent infection. This infection after treatment can either clear (no longer detected) or persist (still detected at follow up). In Table 2, there were 6 women with HPV16 infections “before treatment.” How many of these 6 women cleared the infection and how many persisted? The denominator for those two categories would be n /6. For incident infections the denominator would be 30- 6 = 24.

The same can be said about the biopsied lesions and cytology results. But these would need to be included in a separate table or removed. The purpose of your paper was to assess how treatment of CIN2+ effected the natural history of HPV and not how treatment effected the lesions.

  1. Lines 137-144. Need to be updated based on clearance of HR-HPV infections. This would then assess the association between treatment and clearance of HPV in the same statistical model. Logistic regression, clearance of HPV (outcome) = treatment (exposure). It seems like this was done in line 144 but making sure that it was type-specific clearance of those HR-HPV infections. Fine to group them in the model as long as it is clearance of any HR-HPV infections even if they persisted with a different HR-HPV type.
  2. Lines 137-144 clarify how the 60% and 32% relative decrease in HR-HPV prevalence was calculated? What statistical methods were used?
  3. Lines 150-155. What proportion of women had a persistent infection in the two groups? Did you run univariate analyses or multivariable? Unlikely that there are enough events to be able to run multivariable models with the sample size being so small and the models are being overfit.
  4. The conclusion regarding significant difference in clearance between those with and without treatment are not true given not a statistical difference and confidence intervals including 1.0.
  5. Another limitation is that 5 women in the “untreated” group appear to have never received a biopsy. The authors need to discuss reasons why these women were not treated for cervical lesions which helps to confirm that the inability to be treated would not be associated with HPV clearance/persistence.

Minor:

  1. What low risk HPV types were detected? Please add these genotypes to the methods section.

Also add references to how HR and PHR HPV types were determined. The HR types listed are standard but have not seen these “PHR” listed before.

Author Response

Interesting article assessing HPV natural history following among women with and without treatment for cervical pre-cancer among women with HIV. There are some limitations with how the data is currently being presented and questions on what statistical methods were used for analyses.

We thank reviewer 2 for their through assessment of our work and their comments.

Major

  1. Table 2 & 3 categories need to be updated to include more columns on the natural history of the infection. An HPV infection or a lesion can clear or persist. We can also have a new (incident) HPV infection. “Before treatment” need to be labeled as a prevalent infection. This infection after treatment can either clear (no longer detected) or persist (still detected at follow up). In Table 2, there were 6 women with HPV16 infections “before treatment.” How many of these 6 women cleared the infection and how many persisted? The denominator for those two categories would be n /6. For incident infections the denominator would be 30- 6 = 24.
    We understand this comment, however, based on the small size of this subset of patients from the cohort, it might not appear relevant to reach this level of details.
    Still, in order to clarify what was measured, we renamed the tables 2 and 3 in order to avoid any confusion between a persistent infection or a newly incident infection.
  2. The same can be said about the biopsied lesions and cytology results. But these would need to be included in a separate table or removed. The purpose of your paper was to assess how treatment of CIN2+ effected the natural history of HPV and not how treatment effected the lesions.
    We agree that how treatment affected the lesions is not the core topic of our paper. We moved the information on lesions, biopsy and pap smear results at the time of treatment indication and at the last visit previously in tables 2 and 3 to supplementary materials.
  3. Lines 137-144. Need to be updated based on clearance of HR-HPV infections. This would then assess the association between treatment and clearance of HPV in the same statistical model. Logistic regression, clearance of HPV (outcome) = treatment (exposure). It seems like this was done in line 144 but making sure that it was type-specific clearance of those HR-HPV infections. Fine to group them in the model as long as it is clearance of any HR-HPV infections even if they persisted with a different HR-HPV type.
  4. Lines 137-144 clarify how the 60% and 32% relative decrease in HR-HPV prevalence was calculated? What statistical methods were used?
    Regarding lines 137-142, we compared the prevalence of HR-HPV at the time of treatment indication and at the time of the last visit using McNemar’s tests for paired nominal data among women who received treatment and women who did not separately. The 60% and 32% decrease mentioned here are only descriptive.
    Regarding lines 143 to 145, this was the result of single variable logistic regression assessing whether treatment had an influence on the prevalence of HR-HPV infection by the end of the follow-up. The reviewer  is right that the way this is phrased can lead to some confusion, hence we made some changes to this paragraph to give a clearer picture of our analysis results (lines 171 to 182).
    “Among women who received treatment, the prevalence of HR-HPV infection was 67% [IC95%, 48% to 82%] at the time of treatment indication and 27% [IC95%, 13% to 46%] at the end of follow-up, a 60% relative decrease of HR-HPV infection prevalence compared to the baseline (p value = 0.001) (Table 2). Among women who did not receive treatment, the HR-HPV prevalence was 56% [IC95%, 20% to 70%] at baseline versus 38% [IC95%, 16% to 65%] at the end of follow-up, a 32% relative decrease, which was not statistically significant (p value = 0.257) (Table 3). There was no significant association between prevalence of HR-HPV infection by the end of the follow-up and the fact that women had received treatment (OR = 0.6 [IC95, 0.2 to 2.2], p value = 0.449).”
  5. Lines 150-155. What proportion of women had a persistent infection in the two groups? Did you run univariate analyses or multivariable? Unlikely that there are enough events to be able to run multivariable models with the sample size being so small and the models are being overfit.
    We conducted single variable analysis adjusted for treatment, meaning that the logistic regression model was follow: HR-HP infection explained by variable X adjusted for treatment.
  6. The conclusion regarding significant difference in clearance between those with and without treatment are not true given not a statistical difference and confidence intervals including 1.0.
    We do agree with the reviewer that there was no statistical significance difference in the decrease of HR-HPV prevalence depending on whether women had received treatment. In our manuscript, we had only mentioned in that HR-HPV carriage decreased significantly over time among women who had received treatment. (last sentence of the discussion section)
    “Our analysis is limited by a lack of statistical power, estimated post hoc at 16%. However, we did find that HR-HPV carriage decreased significantly over time among women who had received treatment.”
  7. Another limitation is that 5 women in the “untreated” group appear to have never received a biopsy. The authors need to discuss reasons why these women were not treated for cervical lesions which helps to confirm that the inability to be treated would not be associated with HPV clearance/persistence.
    Treatment was conducted at the discretion of the treating physician. This information is mentioned in the third paragraph of the Material and Methods section (line 97). We also added a sentence explaining that the study team had no influence on whether women would receive treatment or not (lines 98-99).
    The study investigation team had no influence on whether treatment was offered or not to the study participants.”.

Minor:

  1. What low risk HPV types were detected? Please add these genotypes to the methods section
  2. Also add references to how HR and PHR HPV types were determined. The HR types listed are standard but have not seen these “PHR” listed before.
    We thank the reviewer  for noticing this missing information. This is now added to the Materials and Methods section
    Due to the limited evidence for their carcinogenicity, HPV53, HPV 66, HPV 70,  HPV73, HPV 82 were classified as “potentially High Risk HPV” (pHR-HPV) in previous publications related to the papilloV cohort. We clarified this classification in the methods section and added a reference to previous works (lines 101 to 107).
    Genotypes considered high-risk HPV were the following: HPV 16, 18, 31, 33, 35, 39, 45, 51, 52, 56, 58, 59, and 68, while HPV 6, 11, 40, 42, 43 and 44/55 were considered as low-risk HPV genotypes. Due to the limited evidence regarding their potential carcinogenicity, the following genotypes (HPV 53, 66, 70, 73, and 82) were classified as potentially high-risk genotypes (4).”

Round 2

Reviewer 1 Report

I have gone through the response provided by the authors and agree with the corrections authors have included. New MS has a few grammatical errors that authors could easily fix (e.g. line 10 of the abstract “at (the) end of follow”, etc.). Article is now better.

Author Response

Many thanks to reviewer 1 for his feedback.
We corrected minor grammar issues as requested.

Reviewer 2 Report

My initial comments were not addressed. The data is interesting, but the statistical methods are not being described in a enough detail to ensure correct data analysis. The title of the manuscript is "effect of treatment of cervical lesion on clearance of HR-HPV." Therefore the naturally history of HPV needs to be clearly and accurately defined.  If the data is too small to efficiently create the correct categories, then the data cannot answer the scientific questions that were asked. The authors need to define the prevalent infections that went on to clear. For example, 6 women were HPV16 positive at baseline and there was one person that was positive at “last HPV sample.” Is this one women a new infection or a persistent infection?  Additionally, when calculating clearance of HPV the denominator can only include individuals that were “at risk” for the outcome. So the denominator for HR-HPV among those that were treated would be n=20 and among those not treated n=9 because they had a prevalent infection that had the ability to clear or persist.

The authors also ran multivariable logistic regression models but did not provide tables associated with these analysis. No mention of how variables were selected. Too few events to include all of these covariates and models were likely overfit.

Author Response

Reviewer 2: My initial comments were not addressed. The data is interesting, but the statistical methods are not being described in a enough detail to ensure correct data analysis. The title of the manuscript is "effect of treatment of cervical lesion on clearance of HR-HPV." Therefore the naturally history of HPV needs to be clearly and accurately defined.  If the data is too small to efficiently create the correct categories, then the data cannot answer the scientific questions that were asked. The authors need to define the prevalent infections that went on to clear.

For example, 6 women were HPV16 positive at baseline and there was one person that was positive at “last HPV sample.” Is this one women a new infection or a persistent infection?  Additionally, when calculating clearance of HPV the denominator can only include individuals that were “at risk” for the outcome. So the denominator for HR-HPV among those that were treated would be n=20 and among those not treated n=9 because they had a prevalent infection that had the ability to clear or persist.

We understand the reviewer’s point.

However, our design and sample size did not allow us to determine if each HPV subtypes were persistent, incident or cleared at the end of follow-up. Therefore, to improve clarity and to remove any ambiguity, we changed the wording, to mention only the prevalence of HPV infection or HPV carriage by subtypes. We also change the Title of the article as well as the titles of Table 2 and 3.

In the discussion, we also point out this limitation of our work (lines 14-15).

“Also, it should be noted that HPV infections observed at the end of follow-up could be related to persistent infection, but also to new or recurrent infections. It was not possible to differentiate them and this measurement bias cannot be addressed.”

The authors also ran multivariable logistic regression models but did not provide tables associated with these analysis.

No mention of how variables were selected.

Too few events to include all of these covariates and models were likely overfit.

This may be a misunderstanding. We selected a-priori the following variables: age, CD4 cell count, HIV load, time since the patient ART initiation at the time of treatment indication, Nadir CD4 cell count and cervical cytology at the end of the follow-up.

Each model only included one of the above-mentioned variables adjusted on having received treatment.

As reviewer 2 stated, our sample size would not have allowed us to include all these covariates in the same model, and we only performed single variable analyses adjusted on the receipt of cervical lesion treatment.

To improve clarity, the following was added to the last paragraph of the methods section:  “We selected the following variables a priori: age, CD4 cell count, HIV load, time since ART initiation at the time of treatment indication, Nadir CD4 cell count and cervical cytology at the end of the follow-up.”

Additionally, we added the results of these analyses as a supplementary table (Appendix Table 3. : Factors associated to the presence of an high-risk HPV infection by the end of the follow up among women presenting with a cervical lesion. Bivariate analyses adjusted on treatment.)

We thank reviewer 2 for his comments and his suggestions that will improve our manuscript.